# Evolution from the plasmon to exciton state in ligand-protected atomically precise gold nanoparticles

Meng Zhou[1], Chenjie Zeng[1], Yuxiang Chen[1], Shuo Zhao[1], Matthew Y. Sfeir[2], Manzhou Zhu[3] & Rongchao Jin[1]

The evolution from the metallic (or plasmonic) to molecular state in metal nanoparticles constitutes a central question in nanoscience research because of its importance in revealing the origin of metallic bonding and offering fundamental insights into the birth of surface plasmon resonance. Previous research has not been able to probe the transition due to the unavailability of atomically precise nanoparticles in the 1–3 nm size regime. Herein, we investigate the transition by performing ultrafast spectroscopic studies on atomically precise thiolate-protected $Au_{25}$, $Au_{38}$, $Au_{144}$, $Au_{333}$, $Au_{\sim 520}$ and $Au_{\sim 940}$ nanoparticles. Our results clearly map out three distinct states: metallic (size larger than $Au_{333}$, that is, larger than 2.3 nm), transition regime (between $Au_{333}$ and $Au_{144}$, that is, 2.3–1.7 nm) and non-metallic or excitonic state (smaller than $Au_{144}$, that is, smaller than 1.7 nm). The transition also impacts the catalytic properties as demonstrated in both carbon monoxide oxidation and electrocatalytic oxidation of alcohol.

[1] Department of Chemistry, Carnegie Mellon University, Pittsburgh, Pennsylvania 15213, USA. [2] Center for Functional Nanomaterials, Brookhaven National Laboratory, Upton, New York 11973, USA. [3] Department of Chemistry and Center for Atomic Engineering of Advanced Materials, Anhui University, Hefei, Anhui 230601, China. Correspondence and requests for materials should be addressed to R.J. (email: rongchao@andrew.cmu.edu).

Plasmonic metal nanoparticles have found a wide range of applications in nanoantennas, photochemical reactions and solar cells, to name a few[1–4]. A central question in metal nanoparticle research pertains to the evolution from the metallic to molecular state[5]. Spherical gold nanoparticles with diameters between 5 and 100 nm are well known to exhibit a distinct surface plasmon resonance (SPR) between 520–570 nm (wavelength) depending on the size. With decreasing size (below ∼5 nm), the SPR starts to be dampened and blueshifted, and eventually disappears for ultrasmall particles (<2 nm in diameter). The ultrasmall gold nanoparticles in the quantum size regime possess discrete electron energy levels and show molecular-like behaviour such as single-electron transitions or excitons[6], in contrast to the collective excitation behaviour in the metallic state. Mapping out the precise evolution from the metallic (or plasmonic) state to the molecular excitonic state is of major importance[5,7] because it not only reveals the origin of metallic bonding but also offers fundamental insights into the birth of SPR. However, the precise transition and its effects on the particles' properties (for example, catalysis) still remain unclear. In recent years, significant advances have been achieved in the wet chemical synthesis of atomically precise nanoparticles or nanoclusters[8,9], which has opened up new opportunities for fundamental studies at the unprecedented level of atomic precision.

In terms of electronic excitation and relaxation, the behaviour of metallic nanoparticles (for example, 5–100 nm) has been extensively studied[4,10–14]. After absorbing light, electrons in the conduction band are heated up to a very high electronic temperature and then thermalized to reach a new equilibrium in a very short time via electron–electron scattering (on the order of 100 fs); meanwhile, the energy flows from electrons to the ion lattice through electron–phonon coupling (1–5 ps), and finally, the energy is dissipated into the environment through heat transfer and diffusion (10–100 ps)[4]. For ultrasmall non-metallic nanoclusters such as $Au_{25}$, the emergence of discrete states and HOMO-LUMO gap gives rise to single-electron excitations (that is, excitons) and thus long-lived excited states[15–17]. The transition from metallic to excitonic state is reflected in steady-state optical spectra of nanoparticles/nanoclusters[7,8,18], excited state lifetime[19,20], electron and phonon dynamics[21–24], and nonlinear optical properties[21,25]. A characteristic feature of the metallic-state particles is that the initial electron temperature is highly dependent on the pump laser intensity, and the electron–phonon coupling time exhibits high sensitivity to the pump power[4,26,27]. Molecular-like gold nanoclusters, on the other hand, exhibit power-insensitive electron dynamics[16]. Therefore, the electron dynamics measured at different pump powers constitute a distinct signature that differentiates plasmonic and excitonic gold nanoparticles due to the evolution in electronic mobility and screening interaction. The investigation on the transition requires single-sized nanoparticles in the 1–5 nm regime, but it had long been a major challenge to achieve atomic monodispersity until recently[8,28,29].

Here, we utilize the atomically precise gold nanoparticles in the range of 1.0–3.5 nm (including $Au_{25}$, $Au_{38}$, $Au_{144}$, $Au_{333}$, $Au_{\sim520}$ and $Au_{\sim940}$) to investigate the grand transition from the metallic to excitonic state by femtosecond transient absorption spectroscopy as well as the impact of the transition on catalytic properties. By directly probing the electron–phonon coupling in these gold nanoparticles, we explicitly map out that the metallic to molecular state transition occurs between 2.3 nm ($Au_{333}$) and 1.7 nm ($Au_{144}$). The $Au_{333}$ nanocluster exhibits both molecular and plasmonic behaviour and is thus intermediate between the typical molecular state (for example, $Au_{144}$) and the typical metallic state (for example, $Au_{\sim520}$). This transition is also discovered to be coincident with the trend of catalytic activity in the oxidation of carbon monoxide (CO) and electrocatalytic oxidation of alcohol.

## Results

**Characterization.** The syntheses of different sized gold nanoparticles (that is, $Au_{25}$, $Au_{38}$, $Au_{144}$, $Au_{333}$ $Au_{\sim520}$ and $Au_{\sim940}$) were based on size-focusing protocols (see the 'Methods' section). All the nanoparticles were protected by the same type of thiolate (that is, phenylethanethiolate). Figure 1a shows the matrix-assisted laser desorption/ionization mass spectroscopy of the nanoparticles, with their masses being 7, 11, 34, 74, 115 and 200 k (where $k = 1,000$). The precise masses were determined by electrospray mass spectrometry (except for the 115 and 200 k particles) and accordingly the formulas were assigned (Supplementary Note 1 and Supplementary Fig. 1). It is interesting to note that the progression of magic numbers in $Au_{144}$, $Au_{333}$, $Au_{\sim520}$ and $Au_{\sim940}$ magic sizes seems to correlate with the closed-shell icosahedral or cuboctahedral clusters (that is, 147, 309, 561 and 923 for four, five, six and seven shells, respectively)[31,32]. This close relationship indicates a shell-by-shell growth pattern in these magic sizes. The sizes of the four nanoclusters are 1.7, 2.3, 2.9 and 3.5 nm, respectively, according to a relation of $0.578 \times (x–1)$, with $x$ representing the number of shells and 0.578 nm representing twice the Au-Au bond length, whereas the $Au_{25}$ is 1.0 nm and $Au_{38}$ is 1.2 nm according to their X-ray structures[8].

Figure 1b shows the optical absorption spectra of $Au_{25}$, $Au_{38}$ and $Au_{144}$, which possess multiple bands and are characteristic of molecular-state particles; no SPR peak can be observed. The $Au_{333}$, $Au_{\sim520}$ and $Au_{\sim940}$ all exhibit a prominent SPR peak at 540, 525 and 525 nm, respectively. One can see that as the size decreases, the SPR peak becomes broadened (for example, the trend from $Au_{\sim940}$ and $Au_{\sim520}$, Fig. 1b). In contrast, the peak of $Au_{333}$ is more prominent compared with the larger counterparts, which differs from the trend due to the co-existence of molecular and plasmonic behaviour (*vide infra*). While the emergence of the SPR peak is one of the criteria to differentiate metallic and non-metallic gold nanoparticles, ambiguity may exist and it is of critical importance to investigate the electron dynamics to determine the precise onset of metallic behaviour with decreasing size.

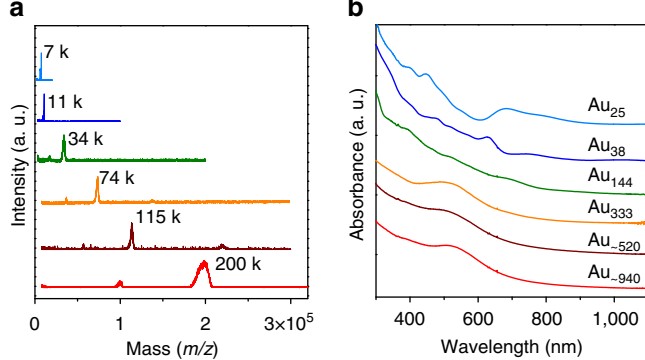

**Figure 1 | Characterization of different-sized gold nanoparticles.**
(**a**) Matrix-assisted laser desorption/ionization mass spectra of the nanoparticles; note that the small peaks at 17.2 k (for the 34 k species), 36.9 k (for the 74 k species), 57.0 k (for the 115 k species) and 100.0 k (for the 200 k species) are doubly charged particles, and the small peak at 220 k (for the 115 k species) is a dimer. (**b**) Steady-state ultraviolet–visible spectra of the nanoparticles.

**Ultrafast electron dynamics.** To probe the electron dynamics of gold nanoparticles, we performed pump power-dependent transient absorption spectroscopy experiments on these gold nanoparticles. As a comparison, the electron dynamics of colloidal gold nanoparticles (diameter: 13 ± 1.5 nm; abbreviated as Au 13 nm NPs) were also included. For gold nanoparticles with the SPR peak, we chose the 480 nm pump wavelength to selectively excite the *d* band electrons (note: *d* to *sp* band transition threshold at ∼2.4 eV, corresponding to 520 nm)[33]. The transient absorption spectra of gold nanoparticles with the SPR peak exhibit similar spectral features, that is, a strong negative bleaching signal due to the SPR and positive wings on both sides of the bleach (Supplementary Fig. 2). Figure 2a–c show the electron dynamics of Au$_{\sim 940}$, Au$_{\sim 520}$ and Au$_{333}$ monitored at the maximum of SPR position with varying pump fluences. The electron dynamics of plasmonic nanoparticles can be separated into three processes: electron–electron scattering (on the order of 100 fs, fast rise), electron–phonon coupling (1–5 ps, fast decay) and heat dissipation into the environment (10–100 ps, slow decay; Supplementary Fig. 3). The pump power-dependent electron dynamics of Au 13 nm NPs are shown in Supplementary Fig. 4a and Supplementary Note 2. As the pump fluence increases, the electron–phonon coupling (abbreviated as e–p coupling) slows down for plasmonic gold nanoparticles. The e–p coupling time increases more dramatically with increasing pump fluences for larger gold nanoparticles (Fig. 2a–c).

The Au$_{144}$, on the other hand, exhibits excited state absorption (ESA) at 600 and 750 nm after excitation at 490 nm, as well as ground state bleaching (GSB) around 460, 515, 560 and 670 nm (Supplementary Fig. 5a). In <20 ps, kinetic traces at all wavelengths decay to zero. The kinetic trace of the GSB at 460 nm was monitored as a function of pump fluence because of its smaller overlap with ESA compared with other wavelengths (Supplementary Fig. 5b). It is interesting to see that as the pump fluence increases from 80 to 600 μJ cm$^{-2}$, the normalized kinetics remain the same (Fig. 2d), which is similar to the

smaller gold nanoclusters[16]. The completely power-independent electron dynamics suggest that Au$_{144}$ (1.7 nm) is a typical non-metallic nanoparticle, rather than in the metallic state[24]. The discrepancy between our results and that of Yi *et al.*[24] is due to their choice of the 525 nm probe wavelength to show the electron dynamics as a function of pump power. We point out that the kinetic trace at 525 nm strongly overlaps with ESA (Supplementary Fig. 5 and Supplementary Note 3), thus it underestimates the decay time. By comparing electron dynamics at different excitation wavelengths, we have further confirmed the power-independent electron dynamics in Au$_{144}$ (Supplementary Figs 6 and 7). Compared with smaller gold nanoclusters such as Au$_{25}$ and Au$_{38}$ (Supplementary Fig. 8 and Supplementary Note 4), the electron dynamics of Au$_{144}$ show a much shorter lifetime (3 ps), which indicates a very small bandgap of Au$_{144}$ compared with Au$_{38}$ ($E_g = 0.9$ eV) and Au$_{25}$ ($E_g = 1.3$ eV)[8]. Overall, both the steady-state spectrum and ultrafast electron dynamics indicate that Au$_{144}$ is in the molecular state, rather than the metallic state. It is worth noting that a similar-sized Au$_{133}$(SR)$_{52}$ nanocluster, which consists of multiple shells, is in the molecular state[5] as in the case of Au$_{144}$(SR)$_{60}$. In contrast, weak power dependence in electron dynamics of Au$_{333}$ starts to manifest at both the SPR peak and other probe wavelengths (Supplementary Fig. 9).

The electron dynamics of metallic-state gold nanoparticles is explained by the well-established two-temperature model[34–36], in which the energy exchange rate between electron and phonon follows,

$$C_e(T_e)\frac{dT_e}{dt} = -g(T_e - T_l) \qquad (1)$$

$$C_l\frac{dT_l}{dt} = g(T_e - T_l) \qquad (2)$$

where $T_e$ and $T_l$ are the temperatures of electron and lattice, respectively, $C_e(T_e)$ is the heat capacity of electrons as a function of $T_e$, $C_l$ is the heat capacity of the lattice and $g$ is

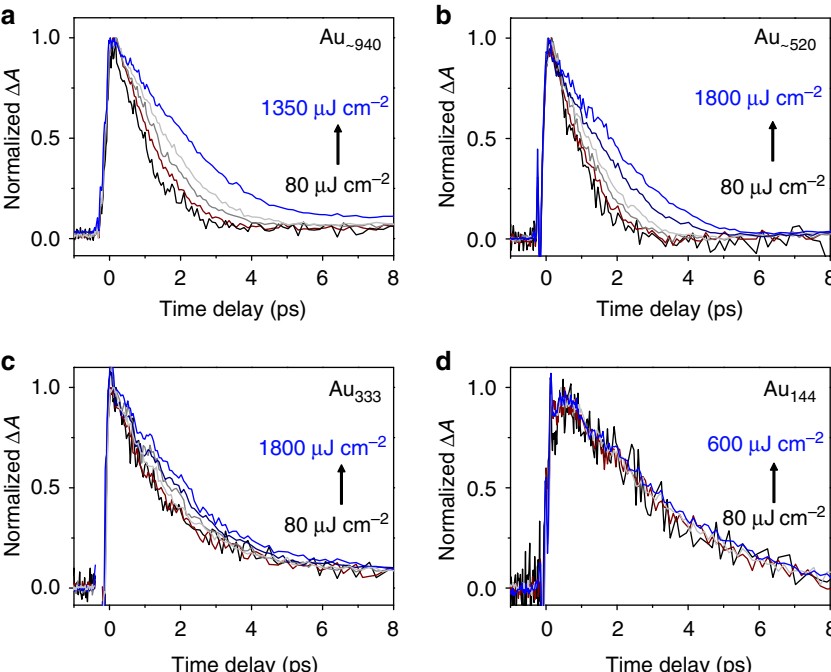

**Figure 2 | Pump power-dependent electron dynamics of gold nanoparticles.** Normalized decay kinetics as a function of pump fluence for (**a**) Au$_{\sim 940}$, (**b**) Au$_{\sim 520}$, (**c**) Au$_{333}$ and (**d**) Au$_{144}$; the kinetics at the maximum of SPR peak of each gold nanoparticle was monitored; for Au$_{144}$, GSB ∼460 nm was monitored.

the electron–phonon coupling constant. Since $C_e$ is dependent on the electron temperature, the energy exchange between electron and phonon (that is, the electron–phonon coupling) is dependent on the fluence of pump pulse. The electron dynamics of plasmonic gold nanoparticles can be well fitted by a theoretical function (Supplementary Fig. 4 and Supplementary Note 2) convoluted by instrument response. It can be seen that the e–p coupling time constant increases linearly with the pump fluence for Au 13 nm NPs, $Au_{\sim 940}$ and $Au_{\sim 520}$ (Fig. 3a). However, $Au_{333}$ exhibits only a weak power dependence while $Au_{144}$ loses power dependence completely (Fig. 3a,b). At low excitation level, the relaxation time of metallic gold nanoparticles can be given by $\gamma(T_0 + \Delta T)/g$, where $T_0$ is the initial electron temperature without pump pulse and $\Delta T$ is the temperature increased by the pump pulse[37,38]. Therefore, the linear fit of the pump fluence-dependent e–p coupling time constants gives the intercept (that is, the intrinsic e–p coupling time) and the slope, which reflects the e–p coupling strength in transient experiments. For ultrasmall gold nanoclusters, however, increasing the pump pulse fluence can only pump more gold nanoclusters to the excited state, which will not alter their electron dynamics. For $Au_{144}$, the slope is zero (that is, non-metallic) and the intercept represents the electron–hole recombination time. In contrast, for $Au_{\sim 520}$ (2.9 nm), $Au_{\sim 940}$ (3.5 nm) and Au 13 nm NPs both the intercept and slope increase with size (Fig. 3c), which suggests that the larger gold nanoparticle exhibits a smaller e–p coupling constant ($g$) and thus needs a longer time for energy to flow from electrons to the lattice. It is interesting to see that $Au_{333}$ has a smaller slope value but a larger intercept compared with those larger gold nanoparticles. Thus, one can find a parabolic trend of the intercepts of different sized gold nanoparticles (Fig. 3c)—as the size decreases, the e–p coupling first speeds up and then slows down after passing the $Au_{\sim 520}$ size. If we plot the e–p coupling

time and slope as a function of size, the slope versus size can be well fitted by a single exponential function (Fig. 3d, red) while the e–p coupling time shows a point of inflection between 2.3 nm ($Au_{333}$) and 2.9 nm ($Au_{\sim 520}$; Fig. 3d, black).

Interestingly, when one looks into the transient absorption spectra of $Au_{333}$ after pump at 390 nm, two negative transient bleach peaks $\sim 540$ and 480 nm can be observed at all time delays (Fig. 4a), which are not observed in larger gold nanoparticles (Fig. 4b). Negative bleach centered at the SPR position in the transient absorption spectra of large gold nanoparticles arises from the sudden rise of the electron temperature, which is different from the GSB features in small molecular-like nanoclusters that originate from photoexcitation. Therefore, multiple bleach peaks in the transient absorption and relatively slower electron–phonon coupling indicate the presence of both molecular and plasmonic behaviour in $Au_{333}$. Because $Au_{333}$ contains both metallic and molecular behaviour, it cannot be classified as a typical metallic gold nanoparticle. Our results suggest the $Au_{333}$ size as the onset of molecular state as the particle size decreases, since the larger size ($Au_{\sim 520}$) exhibits typical metallic behaviour while the smaller size ($Au_{144}$) exhibits molecular behaviour.

Based on all the above experimental results and analysis, we can now discuss the evolution from metallic to molecular state. Those gold nanoparticles larger than $Au_{333}$ (2.3 nm) behave like typical metals, with their optical features well explained by classical physics: the electrons are heated up after absorbing photons, excess energy flows from electrons to the lattice via e–p coupling and then dissipates into the environment. The $Au_{144}$ (1.7 nm) and smaller gold nanoparticles exhibit molecular-like behaviour that is of quantum mechanical nature: as long as the excitation light has higher energy than the bandgap ($E_g$), electrons can be excited and the relaxation is dominated by radiative and non-radiative relaxations. The gold nanoparticles with sizes

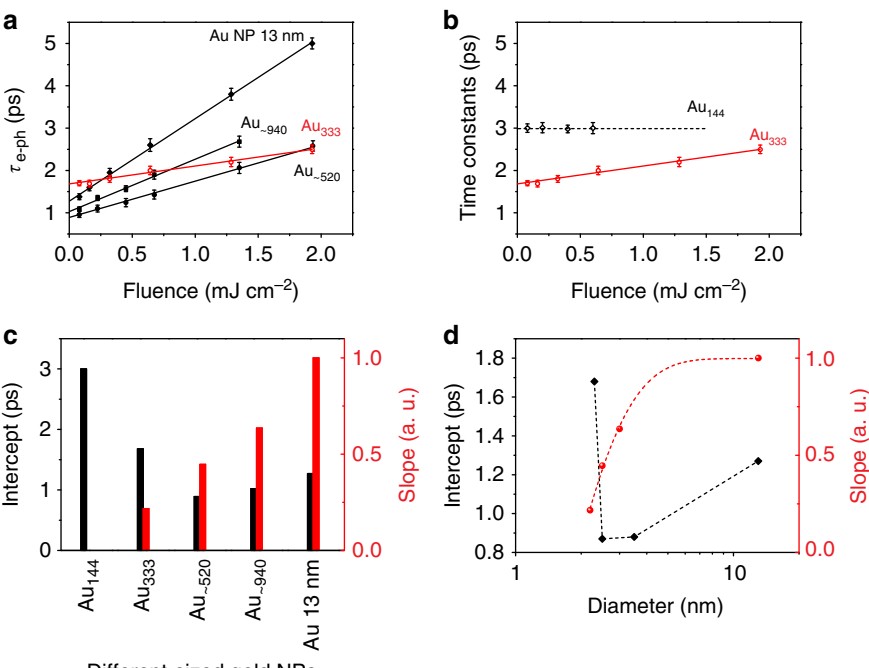

**Figure 3 | Time constants and parameters obtained from the electron dynamics of gold nanoparticles. (a)** The extracted electron–phonon coupling time constants of Au 13 nm NPs, $Au_{\sim 940}$, $Au_{\sim 520}$ and $Au_{333}$ as a function of pump fluence. **(b)** The extracted electron–phonon coupling time constants for $Au_{333}$ and $Au_{144}$ as a function of pump fluence. **(c)** The intercept and slope based on linear fit of the e–p coupling time constant versus pump fluence for all those gold nanoparticles. **(d)** The zero-temperature e–p coupling time constant (black diamond) and slope (red dot) extracted from the fitting as a function of diameter of metallic gold nanoparticles. The error bars in **a,b** represent the s.d.'s of multiple measurements under the same conditions.

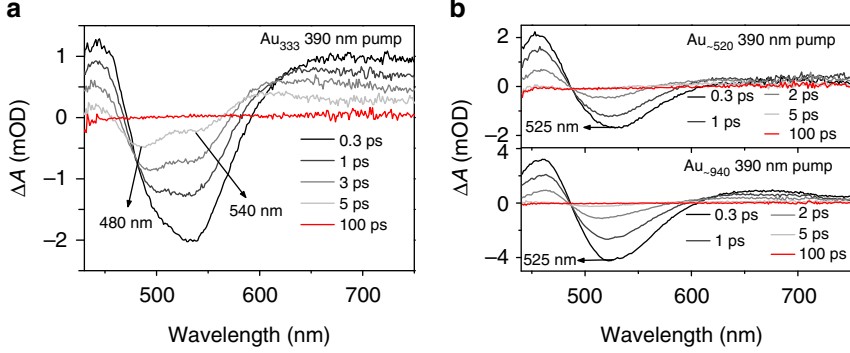

**Figure 4 | Comparison of optical properties of different sized gold nanoparticles. (a)** Transient absorption spectra at different time delays for $Au_{333}$ with 390 nm pump. **(b)** Transient absorption spectra at different time delays for $Au_{\sim 520}$ (upper) and $Au_{\sim 940}$ (lower) with 390 nm pump.

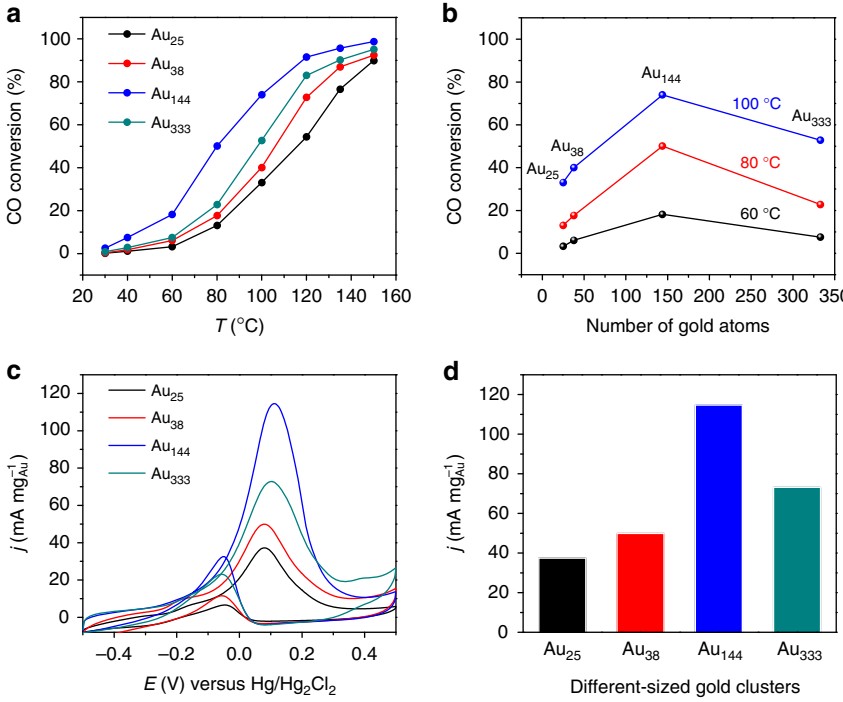

**Figure 5 | The impact of the transition on catalytic CO oxidation and electrochemical oxidation of alcohol. (a)** The light-off curves of different sized nanoparticles, **(b)** a volcano-like trend of size effect, **(c)** CV profiles for different sized nanoparticles in deoxygenated 1M KOH + 1M $C_2H_5OH$ solution, **(d)** comparison of current density of different sized nanoparticles.

between $Au_{144}$ (1.7 nm) and $Au_{333}$ (2.3 nm) constitute the transition regime. When the size is decreased from $Au_{333}$ to $Au_{144}$, the SPR collapses and the bandgap emerges, which mark the onset of complete molecular behaviour.

**Catalysis.** We futher investigated the impact of the transition on catalysis by choosing the CO oxidation and electrocatalytic oxidation of alcohol as the probe reactions. In the CO oxidation reaction, the CO conversion for each size of gold nanoparticles increases with temperature (Fig. 5a), and the size-dependent activity trend (Fig. 5b) indeed coincides with the transition from non-metallic to metallic state, with the most active size within the transition regime (that is, between $Au_{144}$ and $Au_{333}$). Previous work has reported various trends of nanogold-catalyzed CO oxidation[39–42], but the precise size dependence was not clear due to the inherent polydispersity of those nanocatalysts. Different

factors were invoked previously to explain the size dependence, with the highest activity at the size that gives the longest interfacial perimeter distance between gold and oxide support[39], or the size that has more low-coordinated gold atoms[40,41]. Alternative to these geometric factors, Goodman and coworkers invoked the quantum size effect on the electronic structure to explain their observed optimal size (2.5–3 nm) for CO oxidation[42]. Our present results clearly indicate that the optimum size lies in the metallic to molecular-state transition regime, rather than all quantum-sized nanoparticles having high activity. The difference of our results from Goodman's work can be attributed to the different morphologies of particles, that is, quasi-spherical particles in our work versus planar (for example, two atomic layer thick) gold particles in Goodman's work[42]. The calculated turnover frequency of $Au_{144}$ is $2 s^{-1}$, which is comparable to the best catalysts reported[43]. Interestingly, the size trend in the electrocatalytic oxidation of alcohol (Fig. 5c,d)

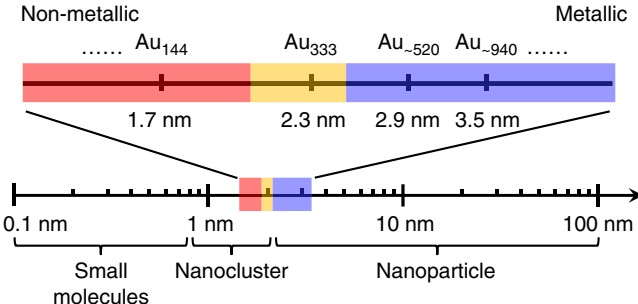

**Figure 6 | Evolution from metallic to non-metallic state in gold nanoparticles.** The size-dependent evolution can be divided into three states: non-metallic or excitonic (red), transition regime (yellow) and metallic or plasmonic (blue).

also reflects the transition from non-metallic to metallic state. Figure 5c shows the cyclic voltammetry (CV) profiles for $Au_{25}$, $Au_{38}$, $Au_{144}$ and $Au_{333}$ (all supported on carbon black), respectively, in which the $Au_{144}$ displays superior ethanol oxidation features with the highest current density in both forward oxidation and reverse oxidation peaks. Figure 5d compares the forward oxidation current density of the four catalysts, where $Au_{144}$ exhibits the best ethanol oxidation activity with a current density of 114.7 mA mg$^{-1}$ Au or 0.36 mA cm$^{-2}$ Au(surface).

## Discussion

Our results explicitly indicate that the transition from metallic to molecular behaviour in gold nanoparticles occurs between $Au_{333}$ and $Au_{144}$ (that is, 2.3–1.7 nm; Fig. 6). $Au_{\sim520}$ and $Au_{\sim940}$ behave like metal, while $Au_{144}$ and smaller particles exhibit molecular-like behaviour. The $Au_{333}$ size exhibits both metallic and molecular behaviour. Based on the optical properties and electron dynamics, gold nanoparticles can be classified into three states: metallic (larger than 2.3 nm), transition regime (between 2.3 and 1.7 nm) and non-metallic (smaller than 1.7 nm). The transition apparently impacts the catalytic properties, as demonstrated in both CO oxidation and electrocatalytic oxidation of alcohol. The determination of the evolution from metallic to molecular gold nanoparticles will open up future exciting opportunities for not only understanding the origin of SPR but also revealing the new properties of metallic nanoparticles in the transition regime.

## Methods

**Synthesis.** All the nanoclusters were synthesized based on the two-step size-focusing methodology[8,30,31]. In the first step, a mixture of $Au_n(SR)_m$ nanoclusters were synthesized by reducing Au(I)SR polymers with NaBH$_4$. Through adjusting the reaction conditions, the $Au_n(SR)_m$ mixtures were controlled to be in a proper size range. In the second step, harsh chemical and thermal conditions (that is, large excess of thiol and high temperature) were applied to select the stable magic size in the size range. The $Au_{25}$, $Au_{38}$, $Au_{144}$, $Au_{333}$ and $Au_{\sim520}$ were synthesized based on the previously developed method[25,30,31], and slight modification of the method led to the next $Au_{\sim940}$ magic size.

**Ultrafast optical measurements.** Femtosecond transient absorption spectroscopy was carried out using a commercial Ti:Sapphire laser system (SpectraPhysics, 800 nm, 100 fs, 3.5 mJ, 1 kHz). Pump pulse was generated using a commercial optical parametric amplifier (LightConversion). A small portion of the laser fundamental was focused into a sapphire plate to produce supercontinuum in the visible region, which overlapped in time and space with the pump. The diameter of the pump beam was 0.75 mm and the pump power was varied between 0.2 and 4.0 mw using neutral density filter. Multiwavelength transient spectra were recorded using dual spectrometers (signal and reference) equipped with array detectors whose data rates exceed the repetition rate of the laser (1 kHz). Solutions of both clusters in 1 mm path length cuvettes were excited by the tunable output of the OPA (pump). All data shown in this manuscript were performed in dilute

solutions using toluene as solvent. During the experiments, all the samples were continuously stirred by a magnetic bar coated by Teflon and ultraviolet–visible absorption remained the same before and after the femtosecond experiments.

**Catalyst preparation and evaluation.** CeO$_2$ (500 mg) support was impregnated by soaking the powders in a 5 ml dichloromethane solution containing different sized gold nanoclusters, that is, $Au_{25}$, $Au_{38}$, $Au_{144}$ and $Au_{333}$, respectively. The ratio of different sized gold nanocluster to the support was tuned to make the gold loading fixed at 0.2 wt%. After impregnated for 24 h, all the nanoclusters were adsorbed onto CeO$_2$ support, as the supernatant became colourless. The as-prepared catalysts were dried at room temperature and no further treatment was performed unless otherwise noted. Catalytic CO oxidation: 100 mg $Au_n$/CeO$_2$ ($n = 25$, 38, 144 and 333, respectively) catalysts were mixed with quartz wool and tested for CO oxidation in a fixed-bed, continuous flow reactor (8 mm i.d.) under ambient pressure. Before CO oxidation test, the as-prepared catalysts were pretreated at 150 °C for 1 h and cooled to room temperature in an O$_2$/He atmosphere. The reaction gas mixture (3% CO, 10% O$_2$ and 87% He) passed through the catalyst bed at a flow rate of 40 ml min$^{-1}$. The products were analysed by an online gas chromatograph (HP 6,890 series GC) equipped with a thermal conductivity detector. The electrochemical measurements were carried out with a three-electrode system on a CHI 620C electrochemical station. A platinum wire and saturated calomel electrode (SCE, Hg/Hg$_2$Cl$_2$) were used as the counter and reference electrodes, respectively. A glassy-carbon electrode (CH Instrument, 3 mm diameter, 0.07065 cm$^2$) was polished with Gamma Alumina powder (CH Instrument). To prepare the working electrode, isopropyl alcohol suspensions of 1 mg of catalyst (total mass) per millilitre were obtained by ultrasonic mixing for about 30 min. Metal-catalyst ink (10 µl) was transferred onto the GC working electrode, followed by 5 µl 0.02 wt% Nafion (diluted from 5 wt% Nafion, Sigma-Aldrich). Before the electrochemical measurements, the electrolyte (1.0 M KOH) was degassed by bubbling nitrogen for 30 min. The CV measurements were processed from −0.5 to 0.5 V for several segments to achieve a stable voltammetry profile. The alcohol oxidation performances were evaluated in 1.0 M KOH solutions with 1.0 M ethanol. The CV scan rate was 20 mVs$^{-1}$.

**Data availability.** All relevant data are available from the corresponding author on request.

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

## Acknowledgements

R.J. thanks financial support by the Air Force Office of Scientific Research under AFOSR Award No. FA9550-11-1-9999 (FA9550-11-1-0147) and the Camille Dreyfus Teacher-Scholar Awards Program. M.Z. is supported by NSFC (21372006, U1532141, 21631001), the Ministry of Education and the Education Department of Anhui Province. Transient optical measurements were carried out at the Centre for Functional Nanomaterials, Brookhaven National Laboratory, which is supported by the US Department of Energy, Office of Basic Energy Sciences, under Contract No. DE-AC02-98CH10886.

## Author contributions

M. Zhou and C. Zeng contributed equally. M. Zhou and M. Y. Sfeir performed fs measurements and data analysis, C. Zeng synthesized the nanoparticles, Y. Chen and S. Zhao carried out the catalytic tests, M. Zhu and R. Jin were responsible for the design of the project. All authors contributed to the writing of the manuscript.

## Additional information

**Competing financial interests:** The authors declare no competing financial interests.

