## [Peer Review File · Nature Communications]

Reviewers' comments:

Reviewer #1 (Remarks to the Author):

This paper describes that the electronic properties of Au particles changes from molecular to metallic states at the boundary of around 2.3 nm (Au₃₃₃ nanoparticle) according to UV-vis, electron dynamics, and transient absorption spectroscopy. This behavior was also observed in catalytic reactions with Au particles on CeO₂: CO oxidation and electrochemical oxidation. Au₁₄₄/CeO₂ having both metallic and molecular features showed higher catalytic activity than smaller (Au₂₅, Au₃₈, molecular state) and larger (Au₃₃₃, metallic state) Au particles. I am wondering the effect of thiolate ligands, because thiolate-protected Au clusters have Au(I) site at the shell and the fraction Au(I) may change the size of clusters. Recently, supported single Au atom catalysts have been reported to be active for CO oxidation (e.g., ACS Catal. 2015, 6249). I am interested in comparison of reaction rates of Au₁₄₄/CeO₂ and Au₁/CeO₂ normalized by surface Au atom. These points will be clear, I feel this paper is very interesting and can be accepted. I am not the expert for electron dynamics, so I will follow the opinions made by other reviewers in the field of electron dynamics.

Reviewer #2 (Remarks to the Author):

Transition from molecular to metallic energetics was probed by ultrafast spectroscopy analysis in ligand stabilized gold nanoclusters. This is a very important question to address in this fast growing field and has broad implications in chemistry, physics and materials science. The authors used the same stabilizing ligand to synthesize a series of gold clusters with 'atomic' precision, at least for the smaller ones. Previous related studies are limited by the size distribution and incoherence in ligand/S-bonding, both addressed in this report. The qualitative agreements and discrepancies with those in literature (i.e. by Goodson, Knapenberger as cited among others) will likely stimulate discussions and future work. I would recommend publication after some minor revisions.

Discussions on why such large clusters like Au₁₄₄ with multiple shells is nonmetallic could provide physical insights for generalization and broaden the scope of the work.

I am not sure if there were sufficient evidence for the ground state bleaching at long wavelength (560 & 670, page 4 & Fig SI-5). explanations?

What's the governing factors for the 3 ps time constant of the Au₁₄₄ cluster? Can one use it to predict other nonmetallic clusters?

Scan rate in CV should be provided. The peak potentials should be compared to other catalysts under the same alkaline conditions.

I will be curious to see the catalytic efficiency normalized by outer shell Au atoms. By weight is acceptable though.

Reviewer #3 (Remarks to the Author):

This manuscript reports studies of the ultrafast electron dynamics and catalytic activity of monodisperse, ligand-stabilized, colloidal gold clusters with sizes from 25 to approximately 950 atoms.

The electron dynamics show a qualitative change from molecule-like for small clusters (with bleaching at discrete transition energies, excited-state absorption, and single decay times) to metal-like for large clusters (with a broadened and shifted plasmon resonance and an electron cooling time that depends on the energy of the exciting pulse). Au₃₃₃ exhibit a combination of both features, and are thus describes as being intermediate between the two regimes. This size also coincides with the point where catalytic activity no longer increases with cluster size, but rather begins to decrease.

These results address the question of how plasmonic behavior emerges in gold nanocrystals as their size increases. This, in turn, is an excellent test bed for the greater question of how bulk behavior emerges from the interactions among atoms, a grand challenge in condensed-matter physics and materials science. Although the limiting behaviors have previously been observed, the current manuscript provides a systematic study, enabled by the spectacular ability to synthesize highly monodisperse colloidal gold clusters over a broad range of sizes. The observation of an intermediate size regime, in particular, provides new insight into the microscopic-macroscopic crossover, and should inspire new theoretical studies. I thus consider the manuscript to be novel and important, and deserving of publication in Nature Communications. I have only a few points that the authors should take into consideration.

1. The results are specific to the ligand-stabilized colloidal gold clusters that the authors synthesize. This is important to emphasize, because the quantitative crossover is undoubtedly different in these clusters than in ligand-free gas-phase clusters, or indeed in other types of ligand-stabilized metal clusters. Especially for the smaller clusters, changes in cluster size are accompanied by changes in geometry that may have an impact on their electronic properties and the molecular-metallic crossover. I would suggest that the authors revise the abstract to clarify the types of ligand-stabilized clusters that are studied, and perhaps add the words "ligand-stabilized" and/or "colloidal" to the title.
2. The authors correctly point out the importance of choosing the correct probe wavelength when extracting time constants from transient-absorption measurements, particularly in the case of the Au₁₄₄ clusters. This ambiguity could be avoided by performing a global analysis on the data. If there is indeed a single time constant, it should be able to extract this time constant from a global analysis based on singular-value-decomposition (relatively common in analysis of transient-absorption data), and a more precise estimate of the time constants may be obtained in this way.
3. As a minor point, the manuscript appears to mostly refer to Au₃₃₃ clusters as intermediate between molecular and metallic, but at some points seems to refer to them as metallic. It would be best to be consistent throughout the text.
4. I also note some minor grammatical errors throughout the text. The revised version should be carefully proofread for correct usage.

Response to Editorial and Reviewers' Comments:

Editorial comments:

Author Reply: We have formatted the manuscript according to the journal's checklist.

Response to Reviewers' comments:

Reviewer #1 (Remarks to the Author):

This paper describes that the electronic properties of Au particles changes from molecular to metallic states at the boundary of around 2.3 nm (Au₃₃₃ nanoparticle) according to UV-vis, electron dynamics, and transient absorption spectroscopy. This behavior was also observed in catalytic reactions with Au particles on CeO₂: CO oxidation and electrochemical oxidation. Au₁₄₄/CeO₂ having both metallic and molecular features showed higher catalytic activity than smaller (Au₂₅, Au₃₈, molecular state) and larger (Au₃₃₃, metallic state) Au particles. I am wondering the effect of thiolate ligands, because thiolate-protected Au clusters have Au(I) site at the shell and the fraction Au(I) may change the size of clusters. Recently, supported single Au atom catalysts have been reported to be active for CO oxidation (e.g., ACS Catal. 2015, 6249). I am interested in comparison of reaction rates of Au₁₄₄/CeO₂ and Au₁/CeO₂ normalized by surface Au atom. These points will be clear, I feel this paper is very interesting and can be accepted. I am not the expert for electron dynamics, so I will follow the opinions made by other reviewers in the field of electron dynamics.

Author Reply: We thank the reviewer for the suggestion and have compared the activity. The calculated TOF is 2 s⁻¹, which is comparable to the best catalysts reported in ACS Catal., 2015, 5, 6249. Revision (page 8): "The calculated turnover frequency (TOF) of Au₁₄₄ is 2 s⁻¹, which is comparable to the best catalysts reported."⁴³

New ref 43. Qiao, B., Liu, J., Wang, Y.G., Lin, Q., Liu, X, Wang, A., Li, J., Zhang T. & Liu, J. Highly efficient catalysis of preferential oxidation of CO in H₂-rich stream by gold single-atom catalysts. *ACS Catal.* **5**, 6249-6254 (2015).

Reviewer #2 (Remarks to the Author):

Transition from molecular to metallic energetics was probed by ultrafast spectroscopy analysis in ligand stabilized gold nanoclusters. This is a very important question to address in this fast growing field and has broad implications in chemistry, physics and materials science. The authors used the same stabilizing ligand to synthesize a series of gold clusters with 'atomic' precision, at least for the smaller ones. Previous related studies are limited by the size distribution and incoherence in ligand/S-bonding, both addressed in this report. The qualitative agreements and discrepancies with those in literature (i.e. by Goodson, Knapenberger as cited among others) will likely stimulate discussions and future work. I would recommend publication after some minor revisions.

Discussions on why such large clusters like Au₁₄₄ with multiple shells is nonmetallic could provide physical insights for generalization and broaden the scope of the work.

Author Reply: We thank the reviewer's suggestion and have added a comment in the revised version. Revision (page 5): "It is worth noting that a similar-sized Au₁₃₃(SR)₅₂ nanocluster—which consists of multiple shells, is in the molecular state⁵ as the case of Au₁₄₄(SR)₆₀. The onset of metallic behavior may be dictated by several factors including the size, atomic structure, and the type of surface ligand."

I am not sure if there were sufficient evidence for the ground state bleaching at long wavelength (560 & 670, page 4 & Fig SI-5). explanations?

Author Reply: The ground state bleaching at 670 nm was not chosen because it has strong overlap with ESA, which makes a precise analysis difficult. Experiments at multiple excitation wavelengths have confirmed the power-independent electron dynamics (see Supplementary Figs S6 and S7).

What's the governing factors for the 3 ps time constant of the Au₁₄₄ cluster? Can one use it to predict other nonmetallic clusters?

Author Reply: As the size of the nanocluster increases, generally the bandgap decreases and the lifetime also decreases. The 3 ps time constant can thus be regarded as the carrier recombination time of Au₁₄₄ (bandgap ~0.1 eV). The governing factor should thus be the ultras-small bandgap of Au₁₄₄. For other similar-sized gold nanoclusters, the time constant may vary based on their size and structure.

Scan rate in CV should be provided. The peak potentials should be compared to other catalysts under the same alkaline conditions.

Author Reply: We have added that "The CV scan rate is 20 mV s⁻¹" in the method part (page 11). The main purpose of this manuscript is to compare the catalytic behavior of nanoparticles in the transition regime. It would not be fair to compare with other catalysts since their composition, size, and surface ligand are quite different.

I will be curious to see the catalytic efficiency normalized by outer shell Au atoms. By weight is acceptable though.

Author Reply: The estimated value is 0.36 mA cm⁻². Revision (page 9): "or 0.36 mA cm⁻² Au(surface)".

Reviewer #3 (Remarks to the Author):

This manuscript reports studies of the ultrafast electron dynamics and catalytic activity of monodisperse, ligand-stabilized, colloidal gold clusters with sizes from 25 to approximately 950 atoms. The electron dynamics show a qualitative change from molecule-like for small clusters (with bleaching at discrete transition energies, excited-state absorption, and single decay times) to metal-like for large clusters (with a broadened and shifted plasmon resonance and an electron cooling time that depends on the energy of the exciting pulse). Au₃₃₃ exhibit a combination of both features, and are thus describes as being intermediate between the two regimes. This size also coincides with the point where catalytic activity no longer increases with cluster size, but rather begins to decrease.

These results address the question of how plasmonic behavior emerges in gold nanocrystals as their size increases. This, in turn, is an excellent test bed for the greater question of how bulk behavior emerges from the interactions among atoms, a grand challenge in condensed-matter physics and materials science. Although the limiting behaviors have previously been observed, the current manuscript provides a systematic study, enabled by the spectacular ability to synthesize highly monodisperse colloidal gold clusters over a broad range of sizes. The observation of an intermediate size regime, in particular, provides new insight into the microscopic-macroscopic crossover, and should inspire new theoretical studies. I thus consider the manuscript to be novel and important, and deserving of publication in Nature Communications. I have only a few points that the authors should take into consideration.

1. The results are specific to the ligand-stabilized colloidal gold clusters that the authors synthesize. This is important to emphasize, because the quantitative crossover is undoubtedly different in these clusters than in ligand-free gas-phase clusters, or indeed in other types of ligand-stabilized metal clusters. Especially for the smaller clusters, changes in cluster size are accompanied by changes in geometry that may have an impact on their electronic properties and the molecular-metallic crossover. I would suggest that the authors revise the abstract to clarify the types of ligand-stabilized clusters that are studied, and perhaps add the words "ligand-stabilized" and/or "colloidal" to the title.

Author Reply: We have followed the reviewer's suggestion and indicated the presence of ligands. Revision: In the abstract, we have added "thiolate-protected". In the title, we have corrected as "ligand-protected, atomically precise gold nanoparticles".

2. The authors correctly point out the importance of choosing the correct probe wavelength when extracting time constants from transient-absorption measurements, particularly in the case of the Au₁₄₄ clusters. This ambiguity could be avoided by performing a global analysis on the data. If there is indeed a single time constant, it should be able to extract this time constant from a global analysis based on singular-value-decomposition (relatively common in analysis of transient-absorption data), and a more precise estimate of the time constants may be obtained in this way.

Author Reply: We have added the global analysis (singular-value-decomposition, SVD) of Au₁₄₄ in Figure S5 SI, which indicates that there is indeed only one decay component in Au₁₄₄.

New Supplementary Figure 5 c,d: (c) Left singular vector and the corresponding fit obtained from singular vector decompose (SVD). (d) Right singular vector obtained from SVD, the sharp artifact around 490 nm is due to the pump laser. SVD analysis suggests that there is one decay component for Au144 with time constant of 3.1 ps.

3. As a minor point, the manuscript appears to mostly refer to Au₃₃₃ clusters as intermediate between molecular and metallic, but at some points seems to refer to them as metallic. It would be best to be consistent throughout the text.

Author Reply: We thank the reviewer for the suggestion. Au₃₃₃ contains both metallic and molecular behavior, it cannot be classified as a typical metallic gold nanoparticle.

Revision (page 7): Those Au₃₃₃ (2.3 nm) and larger gold nanoparticles larger than Au₃₃₃ (2.3 nm) behave like typical metals.

4. I also note some minor grammatical errors throughout the text. The revised version should be carefully proofread for correct usage.

Author Reply: We have corrected the grammatical errors in the revised version.

REVIEWERS' COMMENTS:

Reviewer #1 (Remarks to the Author):

The manuscript was revised according to my comment and can be published as it is.

Reviewer #2 (Remarks to the Author):

The authors addressed my concerns directly and indirectly (through question #2 from reviewer 3). No further revision needed.

Reviewer #3 (Remarks to the Author):

The authors have provided complete answers to all of my previous concerns. The manuscript is of high quality and should be published in Nature Communications.